# Effects of the Antimicrobial Peptide Mastoparan X on the Performance, Permeability and Microbiota Populations of Broiler Chickens

**DOI:** 10.3390/ani12243462

**Published:** 2022-12-08

**Authors:** Chunling Zhu, Yilin Bai, Xiaojing Xia, Man Zhang, Xilong Wu, Yundi Wu, Yueyu Bai, Shanqin Liu, Gaiping Zhang, Jianhe Hu, Hanna Fotina, Lei Wang, Xueqin Zhao

**Affiliations:** 1College of Animal Science and Veterinary Medicine, Henan Institute of Science and Technology, Xinxiang 453003, China; 2College of Veterinary Medicine, Northwest A&F University, Yangling 712100, China; 3College of Fisheries Henan Normal University, Henan Normal University, Xinxiang 453003, China; 4State Key Laboratory of Marine Resource Utilization in the South China Sea, School of Biomedical Engineering, Hainan University, Haikou 570100, China; 5Faculty of Veterinary Medicine, Sumy National Agrarian University, 40021 Sumy, Ukraine

**Keywords:** antimicrobial peptide mastoparan X, broiler chickens, performance, intestinal health, immunity

## Abstract

**Simple Summary:**

The results from this study showed that supplementation with the antimicrobial peptide MPX in the broiler diet could effectively improve the performance and intestinal status of broilers, reduce the levels of inflammation-related factors and increase the mRNA levels of tight junction proteins. In addition, a 16S rRNA intestinal microbiota analysis showed that MPX increased the abundance of ceacal content probiotic lactic acid bacteria.

**Abstract:**

Restrictions on antibiotics are driving the search for alternative feed additives to promote gastrointestinal health and development in broiler chicken production. Proteins including antimicrobial peptides can potentially be applied as alternatives to antibiotics and are one of the most promising alternatives. We investigated whether the addition of MPX to the diet affects the production performance, immune function and the intestinal flora of the caecal contents of broiler chickens. One hundred one-day-old chickens were randomly divided into two groups: control (basal diet) and MPX (20 mg/kg) added to the basal diet. The results indicated that dietary supplementation with MPX improved the performance and immune organ index, decreased the feed conversion ratio, increased the villus length, maintained the normal intestinal morphology and reduced the *IL-6* and *LITNF* mRNA expression levels of inflammation-related genes. In addition, MPX increased the mRNA expression of the digestive enzymes *FABP2* and *SLC2A5/GLUT5* and the tight junction proteins *ZO-1*, *Claudin-1*, *Occludin*, *JAM-2* and *MUC2*, maintained the intestinal permeability and regulated the intestinal morphology. Moreover, MPX increased the *CAT*, *HMOX1* and *SOD1* mRNA expression levels of the antioxidant genes. Furthermore, a 16S rRNA microflora analysis indicated that the abundance of *Lactobacillus* and *Lactococcus* in the cecum was increased after addition of MPX at 14 d and 28 d. This study explored the feasibility of using antimicrobial peptides as novel feed additives for broiler chickens and provides a theoretical basis for their application in livestock.

## 1. Introduction

Antibiotics have been widely used in broiler feed as promoters of intestinal health, growth and feed efficiency [1], but the overuse of antibiotics in poultry feed has led to the emergence of antimicrobial resistances in enteric pathogens and an imbalance in the intestinal flora [2]. The intestinal microbiota play an important role in the immune system of and nutrient absorption in chickens [3]. In addition, antibiotic residues in livestock and poultry lead to organ and immunosuppressive diseases in humans [4,5]. Some countries have banned the use of antibiotic growth promoters (AGPs) based on their threat to human and animal health [6,7]. However, the direct restriction of antibiotic use in poultry will inevitably lead to a decrease in production performance [8]. Therefore, new antibiotic substitutes that avoid the problems caused by antibiotics and improve production performance and gut health in chickens are urgently needed.

Antimicrobial peptides (AMP) are a class of small-molecule peptides composed of 15–100 amino acids that form an important part of the body’s innate immune system and can resist the invasion of pathogenic microorganisms [9]. Many types of antimicrobial peptides can be widely obtained from insects, mammals, amphibians, fish, and plants [10]. The biological functions of antimicrobial peptides mainly include antibacterial, antifungal, and antiviral activities which can improve animal performance [11,12]. The sterilization mechanism of antimicrobial peptides mainly involves destroying the bacterial cell membrane, and thus, the development of drug resistance is difficult [13,14]. To kill and reduce the number of pathogenic bacteria to which animals are exposed during the breeding of livestock and poultry, antimicrobial peptides are usually added to the diet or drinking water [15]. Moreover, the addition of antimicrobial peptides to the diet could improve the growth performance and intestinal morphology and regulate the gut microbiota in broiler chickens [16]. MPX is separated from bee venom, contains four positive charges and has three lysine residues and amidated C termini [17]. The peptide hydrophobicity and the electrostatic interactions between the positive charges of the peptides and the negative charges of the phosphate group of cell membrane phospholipids are associated with the action mode of MPX [18]. The bactericidal activity of MPX can be increased by introducing an unnatural amino acid with an octyl side chain via amino acid substitution at positions 1, 8, and 14 [19]. MPX can relieve skin inflammation and testicular inflammation induced by *S. aureus* and LPS in mice [20,21]. Our group’s previous study found that the intraperitoneal injection of MPX results in good antibacterial activity against *E. coli* infection in mice, reduces intestinal inflammation, and improves intestinal barrier function [22]. The disruption of tight junction proteins in the intestinal mucosal barrier results in increased intercellular permeability [23]. However, the effects of MPX added to the diet of broiler chickens remain unknown.

Due to restrictions on the use of antibiotics for broiler chickens, there is an urgent need to find new feed additive alternatives to antibiotics. The aim of this study was to investigate whether dietary supplementation with the antimicrobial peptide MPX affects the growth performance, intestinal health, intestinal integrity, nutrient transporters and cecal microbiota of broiler chickens. This study explored the feasibility of using MPX as a feed additive for broiler chickens and thus provides a theoretical basis for the application of this antimicrobial peptide to poultry.

## 2. Materials and Methods

### 2.1. Ethics Statement

The Animal Ethics Committee of the Henan Institute of Science and Technology (2021HIST025) approved and performed all animal experiments in accordance with the guidelines of the Animal Welfare and Research Ethics Committee.

### 2.2. Peptide Synthesis

Ji er Sheng Hua (Shanghai, China) synthesized and purified MPX (H-INWKGIAAMAKKLL-NH2) at a purity greater than 98%, and this chemical was dissolved in water and stored at −20 °C.

### 2.3. Experimental Animals

One hundred Arbor Acres (AA) broiler chicks (aged 1 day) were obtained from Jiaozuo Poultry Farming Co., Ltd. (Jiaozuo, Henan, China), and raised in a special animal house for SPF chickens. The chicks were divided into the control and MPX treatment groups, with 10 chicks in each replicate and five replicates in each group. The temperature in the house was maintained at 32 °C in the first week, decreased to 22 °C in week 4 via a 2 °C reduction each week, and maintained at 22 °C until the broiler chickens were 42 days of age. The chickens were raised to an age of 14 days with a basal pelleted diet. After 14 days, MPX (20 mg/kg) was added on the top of the feed provided to the chickens until the end of the experiment. During the experimental period, the chickens were provided feed ad libitum and fresh clean water. The experimental period was 42 days. The basal diet used (based on corn-soybean meal) was manufactured by Hunan Pulemei Feed Co., Ltd., (Changsha, China) according to the recommended nutrition standards [24], and to the NY/T33-2004 feeding standard for chickens. The ingredients and nutrient composition of the basal diet are shown in the Table 1.

### 2.4. Production Performance

The weekly body weight (BW) of the broiler chickens was recorded. The quantity of feed and residue was weighed daily. The feed conversion ratio (FCR), average daily feed intake (ADFI) and average daily gain (ADG) were calculated by recording the broiler chicken weights and feed intake. The ADG, ADFI and FCR of broiler chickens at 1–14, 15–28, 29–42 and 1–42 days of age were evaluated.

### 2.5. Dissection and Collection of Viscera and Intestinal Samples

Broiler chickens were euthanized at 28 d and 42 d by cervical dislocation. Five chickens per replicate were killed. The jejunum, ileal and duodenal tissue samples were rinsed with sterile phosphate buffer and immediately frozen at −80 °C for mRNA analysis. The liver, spleen, lung, jejunum and ileum were placed in 4% paraformaldehyde, and to observe the intestinal morphology, hematoxylin eosin (H&E) staining was performed. The jejunum was collected and placed in 2.5% glutaraldehyde for scanning electron microscopy to observe the state of jejunum villi. The cecal contents were collected for microbiota analysis at 28 d and 42 d.

### 2.6. Gene Expression in the Intestine

The primer sequences used in the qRT–PCR assay are shown in Table 2. The mRNA expression levels of *IL-6*, *IFN-γ*, *LITAF*, *ZO-1*, *Claudin-1*, *Occludin*, *JAM-2*, *MUC2*, *AMY2A*, *FABP2*, *SLC15A2/PepT2*, *SLC2A5/GLUT5*, *CAT*, *SOD1*, and *HMOX1* in the jejunum after MPX addition to the diet at 14 d and 28 d were measured by qRT–PCR. An RNA extraction kit (Sorabio, China, R 1200) was used for the extraction of total RNA. An agarose gel was used to verify the RNA integrity. A RevertAid first-strand cDNA synthesis kit (Thermo Scientific, Waltham, MA, USA, K1621) was used to reverse transcribe complementary DNA (cDNA) using 2 μg of total RNA from each sample. Each reaction (10 μL) mainly contained 5 μL of SYBR Green Master Mix (QuantiNova, China, 330500), reverse primer (10 μM, 0.5 μL), forward primer (10 μM, 0.5 μL), ddH_2_O (3.5 μL) and cDNA (0.5 μL). The thermocycling reaction consisted of 95 °C for 2 min and 40 cycles of 95 °C for 20 s and 60 °C for 30 s with the addition of a melting curve analysis. The relative mRNA expression levels were calculated by 2^−ΔΔCt^, and GAPDH was used as a housekeeping gene [25].

### 2.7. Determination of the Immune Organ Index

Five chickens per replicate were killed at 28 d and 42 d, and 10 chickens were placed in each replicate cage. The thymus, spleen and bursa were separated for weighing to calculate the immune organ index. The immune organ index was calculated as follows: Immune organ index (g/kg) = Immune organ weight (g)/Liveweight before slaughter (kg) [1].

### 2.8. Observation of the Intestinal Morphology 

The proximal jejunum and distal ileum specimens were fixed in 4% paraformaldehyde, gradually dehydrated in 50–100% ethanol, embedded in paraffin, sectioned at 4 μm, and stained with hematoxylin and eosin [5]. Tissue sections of the jejunum and ileum were examined using Image-Pro Express 6.0 (Media Cybernetics, Rockville, MD, USA) and a phase contrast microscope (Nikon Eclipse 80i, Nikon Corp., Tokyo, Japan). The main variables were the ratio of the villus height to the crypt depth, crypt depth, and villus height. Each variable for each chicken was measured at least three times.

### 2.9. Microbiome Total DNA Extraction and PCR Amplification of Target Fragments

The microbiota of the chicken cecal contents were analyzed at 14 and 28 d after the addition of MPX. The total DNA extraction method using sodium dodecyl sulfate (SDS) lysed cells was selected as the most appropriate method for microbiome samples, based on various sources [28], and 1.2% agarose gel electrophoresis was performed to determine the quality of extracted DNA. Microbial ribosomal RNA or specific gene fragments as target sequences that can reflect the composition and diversity of bacterial groups according to the conserved regions in the sequence, can be used to design corresponding primers and change the rRNA gene by adding sample-specific barcode sequences. Regions (single or contiguous multiples) or specific gene fragments were amplified by PCR.

### 2.10. Magnetic Bead Purification and Recovery

Twenty-five microliters of the PCR product was added to a 0.8-fold volume of magnetic beads (Vazyme VAHTSTM DNA Clean Beads, Vazyme. N411-01), and the mixture was then thoroughly shaken and suspended on a magnetic stand for 5 min. The supernatant was then carefully aspirated with a pipette. Subsequently, 20 μL of washing solution was added, the mixture was shaken to fully suspend the beads, and a magnetic rack was used to adsorb the mixture for 5 min. Then, 200 μL of 80% ethanol was added to the supernatant and carefully aspirated again. To move the magnetic beads to the other side of the PCR tube and fully adsorb them, the tube was inverted on a magnetic rack. To completely evaporate the alcohol and crack the magnetic beads, the supernatant was maintained at room temperature for 5 min. Twenty-five microliters of elution buffer was then added to elute the sample. The PCR tube was placed on the adsorption rack for 5 min, and the contents were then transferred to a clean 1.5-mL centrifuge tube for storage after full absorption [29]. 

### 2.11. Fluorescence-Based Quantification of the Amplified Products

A Quant-iT PicoGreen dsDNA Assay Kit (NovoBiotechnology Co., Ltd., Shanghai, China, P7589) and a microplate reader (BioTek, Winooski, VT, USA, FLx800) were used to quantify the PCR amplification. Each sample was mixed in a corresponding proportion according to the fluorescence-based quantitative results and the sequencing volume requirement of each sample.

### 2.12. High-Throughput Sequencing

The TruSeq Nano DNA LT Library Prep Kit (NP-101-1001) from Illumina was used for 16S rRNA gene sequencing. The library was quality-checked using the Agilent High-Sensitivity DNA Kit, which was used to check the quality of the library using an Agilent Bioanalyzer before sequencing. The qualified library concentration was higher than 2 nM, and the Quant-iT PicoGreen dsDNA Assay Kit was used to quantify the library with the Promega QuantiFluor fluorescence quantitative system [30].

### 2.13. Statistical Analysis

The significance of differences in the control and MPX groups was examined using the t-test when the values showed normal distributions within each treatment group. If they showed non-normal distributions, the Mann–Whitney U test was used to examine the differences between the two groups. Differences were considered significant when the *p*-value was <0.05. All the statistical analysis was performed using GraphPad Prism software (version 8.0, La Jolla, CA, USA).

## 3. Results

### 3.1. Effects of MPX on the Performance of Chickens

The ADG, FCR, ADFI, and mortality rate of broiler chickens in the control and MPX groups at the ages of 1–14 d, 25–28 d, and 29–42 d and throughout the 1–42-day period were recorded. As shown in Table 3, the ADG, ADFI and FCR did not differ between the control and MPX groups at 1–14 d. For days 15–28, no significant differences in the ADFI and FCR were found between the control and MPX groups (*p* > 0.05), whereas the ADG of the control group was lower than that of the MPX group (*p* < 0.05). For the period 29–42 d, the ADG and ADFI of the control group were lower than those of the MPX group (*p* < 0.05), whereas the FCR of the control group was higher than that of the MPX group. Furthermore, the FCR of the MPX group was lower than that of the control group during the overall 42-d period. The body weight (BW) of the MPX group was significantly different from that of the control group at 28 d and 42 d (*p* < 0.05). The morbidity was zero in the control and MPX groups during the 14–42-d period. The above results indicate that MPX improved the growth performance of chickens and that the difference in FCR between the control and MPX groups was significant.

### 3.2. Effects of MPX on the Intestinal Morphology and Structure

We first performed necropsy on the chickens after MPX feeding for 14 and 28 d and observed the intestinal morphology. As shown in Figure 1A, the whole intestine of the control and MPX groups was elastic and glossy. H&E staining was then performed to observe the morphological and structural changes of intestinal villi. As shown in Figure 1B, the intestinal villi were neatly arranged in the MPX group; MPX feeding for 14 d and 28 d increased the villus length and decreased the crypt depth compared with those of the control group. Based on the previous necropsy and H&E staining results, we performed scanning electron microscopy to further observe the structure and morphology of the chicken jejunum and ileum villi after MPX feeding for 28 d. The jejunum and ileum villi and microvilli of the MPX group were all normal and closely arranged, and the intestinal microvilli were relatively uniform (Figure 1C). The above results indicate that MPX could maintain the normal shape and structure of intestinal villi.

Table 4 shows the villus length/crypt depth ratios, crypt depths and villus lengths of the jejunum, duodenum and ileum of 28-day-old chickens. At the ileum level, the difference between the MPX and control groups was not significant (*p* > 0.05). At the jejunum level, the villus length and villus length/crypt depth of the MPX group were markedly higher than those of the control group (*p* < 0.05), whereas the crypt depths of the control group were higher than those of the MPX group (*p* < 0.05). At the duodenum level, the villus length of the MPX group was significantly higher than that of the control group (*p* < 0.05), whereas the crypt depths and villus length/crypt depth ratios did not significantly differ between the control and MPX groups (*p* > 0.05). In addition, H&E staining of the spleen, liver, lung and ileum is shown in Figure 2. Compared with the control group, the addition of MPX to the diet of broiler chickens did not cause pathological damage to the liver, spleen or lung. Furthermore, the addition of MPX to the diet of broiler chickens results in a neater arrangement of the villi.

### 3.3. Effects of MPX on the Immune Organ Index

As shown in Table 5, compared with the control group, the thymus and bursa indices observed in the MPX group were significantly higher on at 14 d (*p* < 0.05). At 28 d, the thymus and bursa indices of the MPX group were significantly higher than those of the control group, whereas the spleen index of the control group was not significantly different from that of the MPX group at 14 and 28 d.

### 3.4. Effects of MPX on Immunological Markers

The expression of the immune factors *IL-6*, *IFN-γ* and *LITAF* in the small intestine was evaluated after MPX supplementation. As shown in Figure 3A, the expression of *IL-6* in the jejunum at 28 d was significantly reduced by the addition of MPX to the diet (Figure 3A, *p* < 0.05). Compared with that of the control group, MPX significantly increased the *IFN-γ* mRNA expression level at 14 d (*p* < 0.05), but the difference was not significant at 28 d (Figure 3B, *p* > 0.05). The expression levels of *LITAF* in the jejunum at 14 d and 28 d are shown in Figure 3C. The results show that the expression of *LITAF* in the MPX group was significantly lower than that in the control group at 28 d (*p* < 0.05); however, no difference was detected at 14 d.

### 3.5. Effects of MPX on Jejunum Barrier-Associated Proteins

*Occludin* mRNA expression in the jejunum at 28 d was significantly higher in the MPX group than in the control group (Figure 4A, *p* < 0.05). *ZO-1* and *Claudin-1* mRNA expression in the jejunum was also significantly increased in the MPX group at 14 d and 28 d (Figure 4B,C, *p* < 0.05). As shown in Figure 3E and Figure 4D, *JAM-2* and *MUC2* mRNA expression in the jejunum was also significantly higher in the MPX group at 14 d (*p* < 0.05). However, the mRNA expression of *JAM-2* at 28 d did not significantly differ between the MPX and control groups (*p* > 0.05).

### 3.6. Effects of MPX on Nutrient Transporters and Digestive Enzymes

Nutrient transcription factors and digestive enzymes play important roles in the broiler chicken intestine. Therefore, the effects of MPX on the mRNA expression of the nutrient transporters and digestive enzymes *AMY2A*, *FABP2*, *SLC15A2/PepT2* and *SLC2A5/GLUT5* in the broiler chicken intestine at 14 d and 28 d were further evaluated by qRT–PCR. As shown in Figure 5A, the difference between the control and MPX groups was not significant at 14 d (*p* > 0.05), whereas the MPX group showed decreased *AMY2A* mRNA expression at 28 d (*p* < 0.05). In Figure 5B, the mRNA expression of *FABP2* in the jejunum was significantly increased in the MPX group at 14 d and 28 d (*p* < 0.01). The mRNA expression of *SLC15A2/PepT2* did not significantly differ between the MPX and control groups at 14 d and 28 d (Figure 5C, *p* > 0.05). The mRNA expression of *SLC2A5/GLUT5* in the jejunum was significantly lower in the control group than in the MPX group at 14 d and 28 d (Figure 5D, *p* < 0.05).

### 3.7. Effects of MPX on the Intestinal Antioxidant Capacity

Excessive free radicals produced by oxidative metabolism can cause tissue and cell damage [31]. Therefore, the mRNA expression of gut antioxidant genes was further evaluated by qRT–PCR. The MPX group showed significantly higher *CAT* mRNA expression at 14 d and 28 d compared with the control group. In addition, the *HMOX1* mRNA expression level in the jejunum of the control group was lower than that of the MPX group at 14 d and 28 d (Figure 6B, *p* < 0.05). The mRNA expression of *SOD1* in the MPX group was higher than that in the control group at 14 d (Figure 6C, *p* < 0.05), whereas the difference at 28 d was not significant between the MPX and control groups (*p* > 0.05).

### 3.8. Effects of MPX on the Cecal Microbial Composition

We performed a 16S rRNA analysis of the cecal contents of broiler chickens to investigate the effects of adding MPX to the broiler chicken feed on the cecal microbiota community. As shown in Figure 7, the heatmap analysis of the flora abundance results showed that the gut microbial composition on different days was significantly altered by the addition of MPX. The abundance of *Lactobacillus*, *Lactococcus* and *Parabacteroides* in the MPX group was significantly increased at 14 d and 28 d. The abundance of *Bacteroides* in the MPX group was significantly increased at 14 d but decreased at 28 d. The abundance of *Clostridium* and *Campylobacter* was increased at 28 d, whereas that of *Shigella* and *Enterococcus* was decreased at 28 d, which indicates that MPX could influence the abundance of the cecal microbiota and maintain gut health.

## 4. Discussion

Antimicrobial peptides are potential substitutes for antibiotics in the poultry industry due to the following advantages: small molecular weight, water solubility, difficulty in developing drug resistance, and regulation of intestinal flora and immune function [32]. However, the growth performance and immunologic function of broiler chickens after supplementation with the antimicrobial peptide MPX have not been tested. The results of this study show that MPX supplementation could increase the ADG and decrease the FCR. MPX increased the villus length of the jejunum and duodenum and thereby changed the intestinal morphology. MPX could affect the development of the bursa and thymus of immune organs, decrease the mRNA expression of inflammatory factors, improve intestinal barrier function by enhancing the mRNA expression of tight junction proteins, influence nutrient transporters and digestive enzymes in the jejunum and improve the intestinal antioxidant capacity by enhancing the mRNA expression of antioxidant genes. Furthermore, MPX could regulate the intestinal flora and increase the abundance of *Lactobacillus* and *Lactococcus*. This study provides the first description of the mode of action of MPX on the performance and immune system of broilers and lays a theoretical foundation for the use of antimicrobial peptides as feed additives to improve the production performance and intestinal morphology and regulate the intestinal flora of broiler chickens.

Cytokines play an important role in the immune response, and the expression of proinflammatory cytokines in the body is abnormal, occurring in response to disease, and causes pathological damage as well as decreases in immune function. Thus, cytokines play an important role in the immune system [33]. Previous studies have found that AMP has an important effect on immunomodulatory activity by activating the MAPK signaling pathway [34,35]. *IFN-*γ is a type II interferon which is considered a proinflammatory cytokine and a pleiotropic cytokine with anti-inflammatory properties, and is a major immune response molecule that can activate immune cells and plays an important role in anti-infection and immunoregulation [36]. Immune cells such as macrophages, T cells and B cells acting on other cells can produce *IL-6*, which is significantly increased during inflammatory infection [37]. This study found that the addition of MPX to broiler chicken diets could significantly increase *IFN-γ* mRNA expression and reduce *IL-6* and *LITAF* mRNA expression in the jejunum at different time points. It is speculated that MPX may inhibit the production of proinflammatory cytokines to a certain extent and thereby reduces intestinal inflammation.

The integrity of the intestinal barrier is maintained by the mucus layer of epithelial cells, including apical proteins and weak junctions, which is fundamental to gut health and homeostasis [38]. The disruption of tight junction proteins in intestinal epithelial cells results in increased intercellular permeability [5]. Tight junction proteins include intracellular scaffolding proteins and transmembrane proteins, such as *Claudin*, *Occludin* and *ZO-1* [39]. Xie et al. found that antimicrobial peptides combined with plant essential oil added to broiler chicken diets as feed additives could significantly increase chicken production performance and increase the mRNA levels of *ZO-1* and *MUC2* [1]. Wickramasuriya et al. (2021) found that oral administration of the antimicrobial peptide *B. subtilis*-cNK-2 could enhance the defense of broiler chickens against coccidial infection, effectively improve the production performance of broiler chickens, and increase the mRNA expression of *Occludin*, *ZO-1*, *MUC-2* and *JAM-2*, which results in enhancements in the intestinal integrity and barrier function [26]. Ali Daneshmand et al. (2020) found that supplementation with the recombinant antimicrobial peptide cLFchimera (20 mg peptide/kg diet) in the diet of chickens could improve NE-related intestinal damage, reduce mortality, and increase the mRNA expression levels of the jejunum tight junction-related proteins *Claudin-1* and *Occludin* [40]. The above studies found that barrier-related proteins, including *Occludin, ZO-1, MUC-2,* and *JAM-2*, play an important role in maintaining intestinal barrier function, and antimicrobial peptides could improve the mRNA levels of these proteins. The results of this study are consistent with those of the abovementioned studies. This study found that the addition of MPX to broiler chicken diets could significantly enhance the mRNA expression of the tight junction-related proteins *Occludin*, *ZO-1*, *Claudin-1* and *JAM-2* at 14 d and 28 d, thereby enhancing the intestinal barrier function of broiler chickens.

The intestinal microbiota play an important role in the immune system and nutrient absorption of chickens [41]. The intestine contains a large number of microorganisms, mainly beneficial flora and symbiotic flora. The immune system and these flora can help the host maintain a balance between tolerance to its own antigens and active immunity to pathogens [42]. In fermented foods and beverages, *Lactobacillus* acts as a probiotic that promotes gut health. Similarly, *Lactococcus* belongs to the *Streptococcus* family due to its production of bacteriocins and organic acids and has a number of beneficial functions [43]. An increase in the abundance of *Clostridium* populations has been related to a higher production of butyric acid [44], which can be absorbed and used as an energy source by colonic epithelial cells [45] and is able to modulate the immune system [46]. *Parabacteroides* is related to intestinal integrity disruption, with a higher abundance of *Parabacteroides* reflecting good intestinal barrier function [47]. The intestinal microbiota community abundance may be correlated with the management of body health, and this correlation may be helpful or otherwise for host animals, such as with *Campylobacter,* which is associated with acute bacterial enteritis [48]. This study found that the addition of MPX to the broiler chicken diet could regulate the diversity of cecal microbes and increase the abundance of the beneficial bacteria *Lactobacillus*, *Lactococcus* and *Parabacteroides* at 14 d and 28 d, increase the abundance of *Clostridium* and *Campylobacter*, and decrease the abundance of *Shigella* and *Enterococcus*. Therefore, this study demonstrates that the addition of MPX to the diet is able to reduce pathogenetic abundance and intestinal mucosa inflammation, increase digestion and nutrient availability for absorption, preserve intestinal health and maintain intestinal homeostasis.

## 5. Conclusions

The results from this study showed that dietary MPX (20 mg/kg) supplementation to chickens could improve the growth performance and immune organ index, increase the villus length, maintain the intestinal structure, and reduce the mRNA expression of inflammation-related genes. In addition, MPX increased digestive enzyme and tight junction protein mRNA expression and improved intestinal barrier function. Furthermore, MPX could affect the microbial community assemblage and exert beneficial effects on the cecal microbiota. These positive effects on intestinal function could contribute to improved weight gain.

## Figures and Tables

**Figure 1 animals-12-03462-f001:**
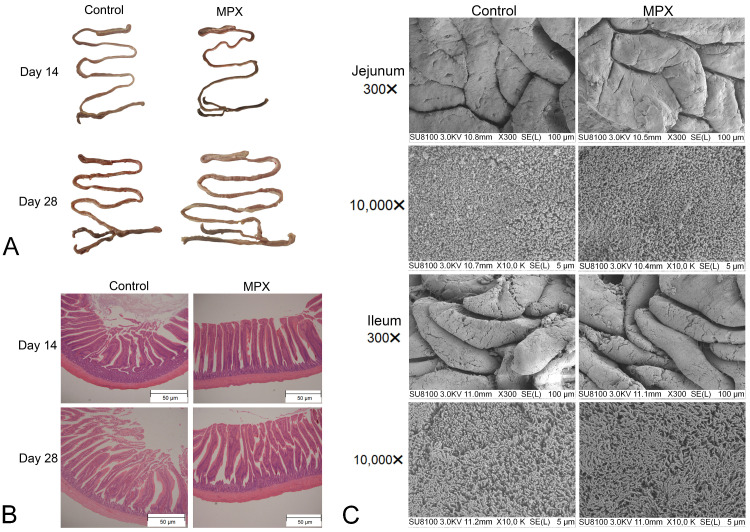
Observation of the intestinal morphology and intestinal villus morphology after MPX supplementation. (**A**) Necropsy observation of the intestinal pathology. (**B**) H&E staining revealed pathological changes in the jejunum. (**C**) Scanning electron microscopy was performed to observe the morphological changes in the villi and microvilli of the jejunum and ileum.

**Figure 2 animals-12-03462-f002:**
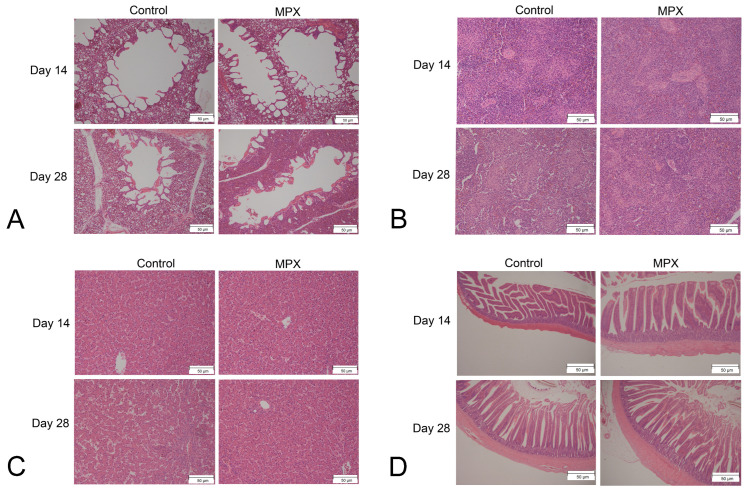
H&E staining was performed to observe the effects of MPX on the viscera and intestine. (**A**) H&E staining of the lung at 14 and 28 d. (**B**) H&E staining of the spleen at 14 and 28 d. (**C**) H&E staining of the liver at 14 and 28 d. (**D**) H&E staining of the ileum at 14 and 28 d.

**Figure 3 animals-12-03462-f003:**
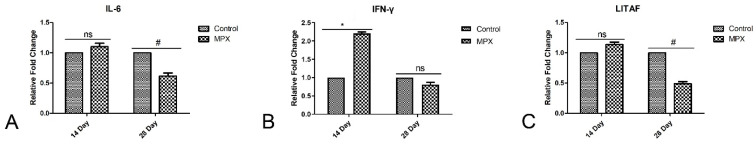
Effects of MPX supplementation on the expression of intestinal immune factors at 14 d and 28 d. (**A**) mRNA expression of *IL-6* at 14 d and 28 d after MPX supplementation. (**B**) mRNA expression of *IFN-γ* at 14 d and 28 d after MPX supplementation. (**C**) mRNA expression of *LITAF* at 14 d and 28 d after MPX supplementation. ns represents no significant difference between control group and MPX group; # represents significant difference between control group and MPX group at 28 d, *p <* 0.05; * represents significant difference between control group and MPX group at 14 d, *p <* 0.05.

**Figure 4 animals-12-03462-f004:**
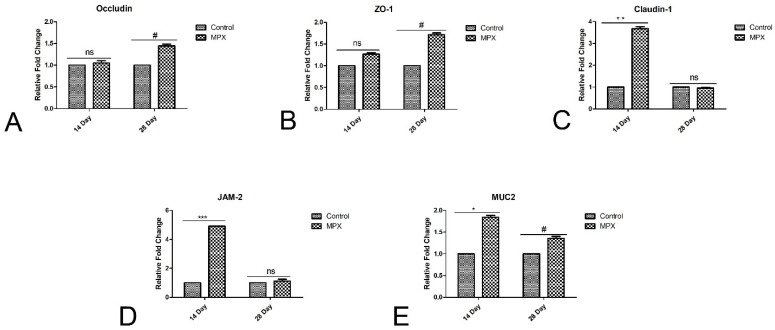
Effects of MPX supplementation on tight junction proteins in broiler chickens at 14 d and 28 d. (**A**) mRNA expression of *Occludin* in chickens at 14 d and 28 d. (**B**) mRNA expression of *ZO-1* in chickens at 14 d and 28 d. (**C**) mRNA expression of *Claudin-1* in chickens at 14 d and 28 d. (**D**) mRNA expression of *JAM-2* in chickens at 14 d and 28 d. (**E**) mRNA expression of *MUC2* in chickens at 14 d and 28 d. ns represents no significant difference between control group and MPX group; # represents significant difference between control group and MPX group at 28 d, *p <* 0.05; * represents significant difference between control group and MPX group at 14 d, *p <* 0.05; ** represents significant difference between control group and MPX group at 14 d, *p <* 0.01; *** represents significant difference between control group and MPX group at 14 d, *p <* 0.001.

**Figure 5 animals-12-03462-f005:**
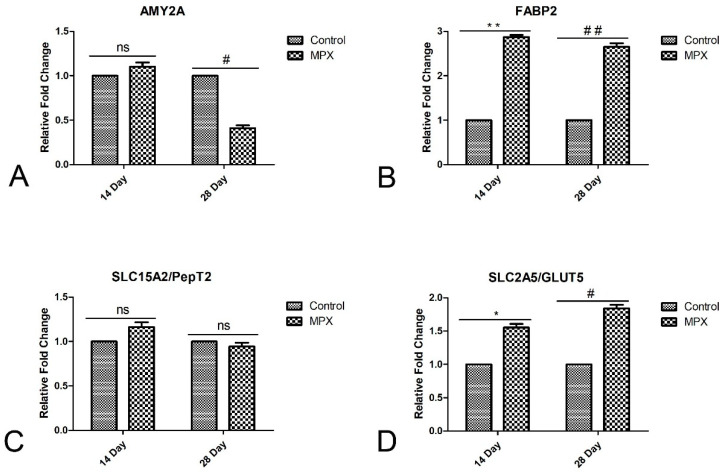
mRNA expression of nutrient transporters and digestive enzymes in the intestine of MPX-supplemented broiler chickens at 14 d and 28 d. (**A**) mRNA expression of *AMY2A* in chickens at 14 d and 28 d. (**B**) mRNA expression of *FABP2* in chickens at 14 d and 28 d. (**C**) mRNA expression of *SLC15A2/PepT2* in chickens at 14 d and 28 d. (**D**) mRNA expression of *SLC2A5/GLUT5* in chickens at 14 d and 28 d. ns represents no significant difference between control group and MPX group; # represents significant difference between control group and MPX group at 28 d, *p <* 0.05; ## represents significant difference between control group and MPX group at 28 d, *p* < 0.01; * represents significant difference between control group and MPX group at 14 d, *p <* 0.05; ** represents significant difference between control group and MPX group at 14 d, *p <* 0.01.

**Figure 6 animals-12-03462-f006:**
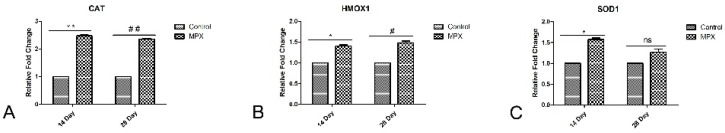
Intestinal antioxidant capacity of MPX-supplemented broiler chickens at 14 d and 28 d. (**A**) mRNA expression of *CAT* in chickens at 14 d and 28 d. (**B**) mRNA expression of *HMOX1* in chickens at 14 d and 28 d. (**C**) mRNA expression of *SOD1* in chickens at 14 d and 28 d. ns represents no significant difference between control group and MPX group; # represents significant difference between control group and MPX group at 28 d, *p <* 0.05; ## represents significant difference between control group and MPX group at 28 d, *p <* 0.01; * represents significant difference between control group and MPX group at 14 d, *p <* 0.05; ** represents significant difference between control group and MPX group at 14 d, *p <* 0.01.

**Figure 7 animals-12-03462-f007:**
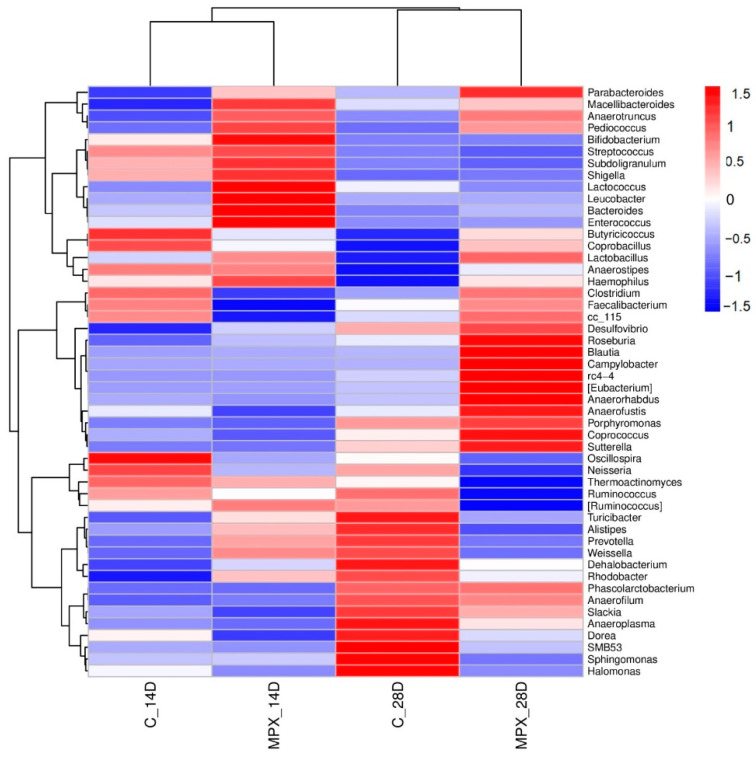
Heatmap analysis of the abundance of intestinal flora.

**Table 1 animals-12-03462-t001:** Nutrient composition of the basal diet (%).

Items	Pre-Starter (Days 1–14)	Starter (Days 15–28)	Finisher (29 Days Onward)
Ingredients			
Corn	54.2	57.2	59.2
Soya bean meal	34	31.5	22.5
Rapeseed meal	5.0	5.0	9.5
Calcium bicarbonate	1.5	1.3	1.3
Mountain flour	1.0	1.2	1.2
Salt	0.3	0.3	0.3
Oil	3.0	2.5	3.0
Wheat bran			2.0
Premix ^1^	1.0	1.0	1.0
Nutrient composition			
Metabolizable energy (MJ/kg)	12.54	12.54	12.96
Crude protein (%)	21.50	21.50	20.00
Total phosphorus (%)	0.68	0.68	0.65
Lys	1.15	1.15	1.03
Met	0.51	0.51	0.41
Met + Cys	0.87	0.87	0.76

^1^ Vitamin Premix provided the following per kg of diets: VA 10,000 IU, VD3 2500 IU, VE, 15 mg; VK3 3 mg; VB1, 0.01 mg; nicotinic acid, 34 mg; calcium pantothenate, 12 mg; folic acid, 0.5 mg; biotin, 0.2 mg; choline chloride, 1200 mg; Fe (as ferrous sulphate), 80 mg; Cu (as copper sulphate), 8 mg; Zn (as zinc sulphate), 80 mg; Mn (as manganese sulphate), 100 mg; I (as potassium iodide), 0.7 mg; and Se (as sodium selenite), 0.3 mg.

**Table 2 animals-12-03462-t002:** List of primers used in the present study.

Gene	Group	Primer Sequence (5′-3′)	Reference
HMOX1	Anti-oxidant gene	F-CTGGAGAAGGGTTGGCTTTCT	[26]
		R-GAAGCTCTGCCTTTGGCTGTA	
CAT		F-ACTGCAAGGCGAAAGTGTTT	[26]
		R-GGCTATGGATGAAGGATGGA	
SOD1		F-ATTACCGGCTTGTCTGATGG	[26]
		R-CCTCCCTTTGCAGTCACATT	
JAM-2	Markers of intestinal integrity	F-AGCCTCAAATGGGATTGGATT	[26]
		R-CATCAACTTGCATTCGCTTCA	
ZO-1		F-GCCTGAATCAAACCCAGCAA	[1]
		R-TATGCGGCGGTAAGGATGAT	
Occludin		F-CCCAGAAGACGCGCAGTAAG	[27]
		R-GCGCGGTCCCAGTAGATG	
MUC2		F-TTCATGATGCCTGCTCTTGTC	[1]
		R-CCGTAGCCTTGGTACATTCTTGT	
Claudin		F-CATACTCCTGGGTCTGGTTGGT	[6]
		R-GACAGCCATCCGCATCTTCT	
FABP2	Nutrient transporters and digestive enzymes	F-CTTGGAAAATAGAGAAAAATGAGAACTATG	[27]
		R-GGCTCCTAACTTTCTTTTCATCACA	
SLC15A2/PepT2		F-CGAAACTCTGTGGCTCCAACT	[27]
		R-CGCTCGCAGAACTCGTTCA	
SLC2A5/GLUT5		F-GGATCAATGCAGTCTTCTACTATGCA	[27]
		R-CACCTATGGACACGGTGACATACT	
AMY2A		F-CACGGGCACCCACTCAAC	[27]
		R-GGCACAGCGGGAAAATCTC	
LITAF	Markers of immune response	F-CCCCTACCCTGTCCCACAA	[27]
		R-TGAGTACTGCGGAGGGTTCAT	
IFN-γ		F-ATCATACTGAGCCAGATTGTTTCG	[1]
		R-TCTTTCACCTTCTTCACGCCAT	
IL-6		F- GTGTGCGAGAACAGCATGGAGA	[5]
		R-CTGGAGAGCTTCGTCAGGCATT	
ACTB	Housekeeping reference genes	F-TGCTGCGCTCGTTGTTGA	[27]
		R-CGTCCCCGGCGAAA	
GAPDH		F-GCTGTGGAGAGATGGCAGAGGT	[5]
		R-ACGGCAGGTCAGGTCAACAACA	

**Table 3 animals-12-03462-t003:** Effects of MPX on the growth performance of broiler chickens at different stages.

Stage	Index	Control	MPX
1 day	BW (g)	44.26 ± 1.03 ^a^	45.13 ± 0.97 ^a^
14 days	BW (g)	235.75 ± 0.86 ^a^	235.45 ± 0.58 ^a^
28 days	BW (g)	684.59 ± 0.49 ^a^	719.57 ± 1.21 ^b^
42 days	BW (g)	1260.97 ± 1.74 ^a^	1322.55 ± 0.53 ^b^
1–14 days	ADG/g	14.73 ± 0.55 ^b^	14.64 ± 0.50 ^b^
	ADFI/g	29.03 ± 0.12 ^a^	28.85 ± 0.58 ^a^
	FCR	1.97 ± 0.08 ^a^	1.98 ± 0.05 ^a^
15–28 days	ADG/g	32.06 ± 1.72 ^b^	34.58 ± 1.35 ^a^
	ADFI/g	53.45 ± 0.58 ^a^	52.28 ± 1.31 ^a^
	FCR	1.67 ± 0.09 ^a^	1.51 ± 0.05 ^a^
29–42 days	ADG/g	41.17 ± 1.26 ^b^	43.07 ± 1.34 ^a^
	ADFI/g	92.67 ± 0.93 ^b^	93.90 ± 0.71 ^a^
	FCR	2.24 ± 0.01 ^a^	2.18 ± 0.03 ^b^
1–42 days	ADG/g	24.87 ± 1.5 ^b^	28.88 ± 2.04 ^a^
	ADFI/g	74.48 ± 1.3 ^a^	76.64 ± 0.77 ^b^
	FCR	2.95 ± 0.09 ^a^	2.7 ± 0.05 ^b^
1–14 days	Mortality (%)	6 (3/50)	4 (2/50)
15–28 days	Mortality (%)	0	0
29–42 days	Mortality (%)	0	0
1–42 days	Mortality (%)	6 (3/50)	4 (2/50)

^a^ represents *p* < 0.05 level difference; ^b^ represents *p* < 0.05 level difference.

**Table 4 animals-12-03462-t004:** Effects of MPX on the intestinal villus index of the ileum, jejunum and duodenum of 28-day-old chickens.

Site	Index	Control	MPX
lleum	Villus length	683.33 ± 9.61 ^a^	681.67 ± 9.61 ^a^
	Crypt depth	156.67 ± 7.09 ^a^	154.33 ± 11.02 ^a^
	Villus length/Crypt depth	4.37 ± 0.16 ^a^	4.43 ± 0.34 ^a^
Jejunum	Villus length	974.33 ± 6.51 ^a^	985.67 ± 9.87 ^b^
	Crypt depth	163.67 ± 8.33 ^a^	151.00 ± 8.54 ^b^
	Villus length/Crypt depth	5.96 ± 0.31 ^a^	6.53 ± 0.32 ^b^
Duodenum	Villus length	1356.00 ± 13.11 ^a^	1423.67 ± 11.06 ^b^
	Crypt depth	255.33 ± 7.09 ^a^	256.33 ± 9.07 ^a^
	Villus length/Crypt depth	5.31 ± 0.20 ^a^	5.56 ± 0.18 ^a^

^a^ represents *p* < 0.05 level difference; ^b^ represents *p* < 0.05 level difference.

**Table 5 animals-12-03462-t005:** Thymic, bursal, and spleen indices of broiler chickens fed a diet supplemented with MPX and measured at 14 d and 28 d.

Stage	Indicator	Control	MPX
Day 14	Thymus	2.01 ± 0.61 ^a^	1.10 ± 0.68 ^b^
	Bursa	2.45 ± 0.18 ^b^	2.95 ± 0.85 ^a^
	Spleen	0.68 ± 0.12 ^a^	0.97 ± 0.30 ^a^
Day 28	Thymus	2.03 ± 0.66 ^b^	2.41 ± 1.05 ^a^
	Bursa	2.27 ± 0.48 ^b^	2.67 ± 0.92 ^a^
	Spleen	1.08 ± 0.25 ^a^	0.91 ± 0.21 ^a^

^a^ represents *p* < 0.05 level difference; ^b^ represents *p* < 0.05 level difference.

## Data Availability

All the data from this study are included in the article, and the corresponding authors can be contacted directly for further queries.

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
