# Peer review of "Effects of the Antimicrobial Peptide Mastoparan X on the Performance, Permeability and Microbiota Populations of Broiler Chickens"

_animals, 2022, doi:10.3390/ani12243462_

Round 1

Reviewer 1 Report

The results presented in the article entitled “The Effect of the Antimicrobial Peptide Mastoparan X on Broiler Chickens” are promising. The search of alternatives to antibiotics used as growth promoters is one of the most important challenges in animal nutrition and the use of antimicrobial peptides is becoming relevant in this field of study. However, the paper needs a deep revision to be publishable in Animals Journal (specially the discussion section).

In this revision I’ve tried to highlight all concepts, mistakes, and sentences which need to be modified or rewritten, and also some suggestions to improve the document.

Line 2: The title should give us more information. I suggest including what improves the AMP (performance, permeability, microbiota populations?) 

Line 26: Delete point

Line 28: You cannot use the word “could”. After the statistical analyses, you must know if a trait is or not improved, if it is a trend…

Line 33: How the AMP regulated the intestinal morphology?

Line 44: zoonotic pathogens

Line 45: decrease in production performance and a deterioration of animal health (include a citation)

Line 59: “not easy to develop drug-resistance“. This is one of the most important characteristics of AMPs. Please, this concept should be more detailed in the introduction (include citations)

Line 61: …water (citation).

Line 64: “MPX separated from bee venom and contained 4 positive charges”. This is the unique sentence describing your product in the introduction. The characteristics of the MPX must be widely described here.

Line 65 (and in all doc): Please, avoid the use of words such as good, bad, more, less, very, few..

Lines 68-79: Your results should not be presented here. Only the aims of the study.

Line 71-72: Revise the format

Line 81: Include the number of ethical authorisation.

Line 87: Clarify “very soluble”

Line 90: What means AA? Arbor Acres (AA)?

Line 95: Please, change the term “normal diet” for commercial or basal diet.

Line 96: The test product was added on top to the feed?

Line 100: Please include a citation for “recommended nutrition standard”. Are you referring to NRC 1994?

Line 104: Please standardize the conversion. In the article is presented as F:G ratio but also as FCR. Personally, I prefer FCR.  

Line 106: Revise the format

Line 109. In the line 96, the concentration of MPX in the feed indicated is 20 ppm, however, in the table you are introducing the additive at 1%. Please, describes the carrier and the % of active substance in it.

Line 109: The table only includes ingredients. The nutritional values of formulation aren’t showed. Values such us Energy, dry matter, fat, or crude protein should be presented.

Line 110: How many samples you used for the analysis of morphology, gene expression, immune index organ, and 16s?

Line 112: and immediately frozen at -80º?

Line 121-125, 154, 166, 171: Indicate the reference number of kits

Line 130: modify the title: “List of primers used in the present study”. Include 5’ and 3’, and also the corresponding citations if the primers were not designed by you. If the primers are house-made, please, describe how you designed them.   

Line 130: You cannot start a sentence with a number. A four percent ….

Line 145: Which is the most appropriate total DNA extraction method (reference)? Did you sequence the 16s?

Line 177: The statistical analyses are not correctly described.

The effects of dietary treatment were analysed by ANOVA using the xxx procedure/program. Was the pen considered as the experimental unit?  What was the model used? Did you analyse your data to look for outliers? How? Smirnoff-Stefansky’s test? When the effect of the dietary treatment was significant, were multiple comparisons of treatment means performed using Tukey test / Fisher’s LSD test / other test? Was the level of significance set at p < 0.05, and trends considered at p < 0.1?

Line 185: (and in all doc): Please use Feed to conversion ratio (FCR)

Line 196: Could you include the mean of body weight per pen in the table? Was the mortality assessed?

Line 219-221: Only in jejunum and villus length in duodenum. In the results section you only should mention the results obtained. This sentence should be in the discussion section.

Line 231: Please, describe the pictures A, B, C, and D in the footnote of S.Figure 1. Why this is not the figure 2?

Line 233-234: This sentence is not a result. It should not appear here.

Line 236: Did you mean that “compared with control group, the thymus, bursa and spleen indices observed in the the MPX group were significantly higher on at 14 d”?

Line 239-240; 252-253, 269-271, 290-291, 306-308: To be discussed in the discussion, not here.

Line 260-261: The results section should not include citations. This sentence should be moved to the introduction.

Line 261-262: You explained it in materials and methods.

Line 315-316: this sentence should be included in the introduction. Not here.

Line 320-321: Only lactobacillus and Lactococcus were statistically increased due to the addition of MPX? The heatmap gives us a lot of information but the reader does not know if the changes are or not statistically significant. Moreover, there are some bacteria populations relevant for poultry production has not taken into account (Clostridium, Enterococcus, Campylobacter, Prevotella…). Moreover, the relative abundance (%) of each bacteria is also relevant. Maybe, the ratio firmicutes/Bacteroidetes could be interesting.

Line 326: The expression “the best” is daring. Real alternatives to antibiotics?

Line 239: It is not true! You just described it in your results! “For the first time, the mode of action of MPX on performance and immune system of broilers have been described”

Line 330-332: This sentence should be moved to the last paragraph of introduction.

Lines 332-343: This paragraph is a summary of results. Maybe it should be included in the abstract but not in the discussion. A discussion about the improve of performance was not include.

Line 348: IFN-g is considered as a pro-inflammatory cytokine. However, it also is classified as a pleiotropic cytokine with anti-inflammatory properties. It should be discussed here.

Line 357-375: In this paragraph you enumerate different articles in which the effects of AMP on intestinal permeability was assessed. However, you do not discuss these results with your results obtained in your study.  

Line 376-388: this paragraph should be rewritten taking into account the changes to be done in the results section.

Line 387: Change the word “probiotic”. The correct term is “beneficial bacteria”. A probiotic is bacterial product administered orally.

Best regards;

Reviewer 2 Report

The topic is interesting and worth of publication but you will first need to make some fairly major changes to the paper (particularly but not limited to the discussion) to satisfy the comments made. Some results are poorly interpreted which affects the quality of the discussion. Please make the discussion straight to the point and DO NOT repeat (overkill) the results in the discussion but explain what the results mean and how all together they support the hypothesis which needs to be stated clearly also. 

Title:

- The title tells NOTHING about the central points of the paper… performance? Gut health? Immunity?? Please improve the title.

L19 reduce

L19 increase

L20 showed

L20 increased

L 23 Proteins? which protein? be more specific

l 25 this sentence is not complete. rewrite. "

 In order to investigate whether addition of MPX to the diet 25 have effects on broiler chickens. "

l 27 showed or indicated

L29 Feed to gain is performance so no need to repeat, or if you wanted to be specific, write the sentence in a better way

L33. You already said "in addition" in L 30 so use something else e.g. Moreover...

L35 was increased where? which segment? indicate please

L43 lead to...

L43 in poultry feed

L45 rewrite this sentence. Language can be improved

L58 "is mainly act" re-write/correct this

L62, "In addition, the addition" use something else than in addition...

L65 "intraperitoneal injection of MPX " to which animal??? specify please

L70 to 79: what is this paragraph exactly?? you already stated the objective of the study in L68 so what is the purpose of the next paragraph?

L97 experimental over? what does this mean? rewrite please

L98 diets were in mash form or pelleted? please specify

L106 rearrange this "the basal diet composition..... in Table 1.." this is NOT production performance!

-Nutritional information? where is the information here? there is diet composition but what nutritional information you are talking about? Where is the analysed composition of the diet?

What is the additive? 1%

-You already mentioned % in the title so no need to have the % sign next to each number in the table!

L110 rewrite

L111 please specify how many birds per replicate were killed. Also, did you not collect caeca for microbiota analysis? Why is it not mentioned?

L112 Please specify which tissue was used for the mRNA/gene expression analysis??? for example AMY2A, was it measured in the jeunual tissue? ileal tissue? specify and give details....

L132 "Chicken were slaughtered at 28 d and 42 d, " you already mentioned this in section 2.5

L132 how many chickens were killed and sampled?? 1 or 2 or 10 per replicate cage or pen? specify!

L138 "to deal" what is meant here?

L144 which section of the digestive tract was used for microbiota analysis? please specify

L155 rewrite this

L179, you only have two groups, so why did you use one-way analysis of variance? why not student t-test? Also, you used a parametric test for all parameters.. Were all parameters normally distributed?? was this checked?

L184 "In order to study the effect of MPX on the performance of broilers " why this sentence is needed?? just start with the results...

L189 "were not difference " rewrite

L221 How was the selection of the sample chosen? was it blinded? was the person performing the analysis aware of the treatments?

L233 "The immune organs play a crucial role in chickens. " what is this sentence doing in the results section????? why you need it here?

L316. You start with intestinal tract... But you analysed caecal contents.. So be specific.

L319 gut microbiota? you mean caecal microbiota?

L326, Please provide a reference to support your claim. Are these really strong arguments for antimicrobial peptides to be the best substitute? " small molecular weight?... good water solubility?

L330. you say it is unknown.. and in the previous sentence you say "expected to be the best substitute" that is the opposite

Where is assessment of growth performance in the aim of the study?

L331 whether effect?? rewrite

L344 what does the first sentence mean?

L386 ceaca is intestine?? That is completley different segments (enviornment, function, flora etc...)

L395 membership? What do you mean?

You start in the discussion by listing your results: You move from L333 "could increase ADG and the mRNA expression" how are these two parameters connected for it to be joined in the same sentence?..

-Better put everything in order and/or category (growth performance parameters together, inflammatory markers together and so on...) For the rest of the discussion, please see first paragraph here in my comments.

Round 2

Reviewer 1 Report

Dear Authors,

The changes done in the manuscript clearly improved the scientific level of the document. However, I’ve detected some mistakes yet, which should be fixed. I used the pdf version 2 (with changes tracking).

Line 28: One hundred day-old chickens were randomly…

Line 45: but the overuse of antibiotics in poultry feed lead the emergence of antimicrobial resistances in enteric pathogens and an imbalance in the intestinal flora.

Line 60: and antiviral activities which can improve animal performance.

Line 105: The test product was added on top to the feed?

Line 106: ad libitum in italics.

Line 107-111: Replace existing paragraph with “The basal diet used (based on corn-soybean meal) was manufactured by Hunan Pulemei Feed Co., Ltd., according to the recommended nutrition standards [24], and to the NY/T33-2004 feeding standard for chickens. The ingredient and nutrient composition of the basal diet are shown in the Table 1.”

Line 108: Table 1: the premix composition1 is not included (as a footnote). Also include the international units for ME (MJ/kg), Crude protein (%), TP (total phosphorus?), and AA in %.

Line 188-189: Replace existing paragraph with “The TruSeq Nano DNA LT Library Prep Kit (NP-101-1001) from Illumina was used for 16S rRNA Gene Sequencing.”

Line 405-407: This paragraph should be rewritten. “An increase of Clostridium populations abundance has been related with a higher production of butyric acid (citation), which can be absorbed and used as an energy source by colonic epithelial [49] and is able to modulate the immune system (citation)”

Line 408: This sentence should be discussed “Parabacteroides is related to intestinal integrity disruption [50]”. What is the relationship with your results? Is it beneficial or not?

Line 411-417: This sentence is too long. “…and decrease the abundance of Shigella and Enterococcus. Therefore, this study demonstrates that the addition of MPX to the diet is able to reduce pathogenetic abundance and intestinal mucosa inflammation, increase digestion and nutrient availability for absorption, preserve intestinal health and maintain intestinal homeostasis.”

Best regards.

Reviewer 2 Report

The manuscript has been improved significantly but there are few and minor comments that need to be taken into consideration.

-What does permeability mean in the title?

l22-the abundance of intestinal probiotic lactic acid bacteria. You analysed ceacal content, so you should be specific. (intestinal can also mean duodenum, jejunum, ileum).

l27: did you analyse caecal contents? or contents from the small intestinal segments?

l79: this study aimed to investigate... then l82: the aim of the study.. why not combining the aims?

l100: what is SPF??

l113: why a quantity???

l104- correct to: with a basal pelleted diet 

l107-108: correct to: The basal pelleted diet were corn- and soybean meal-based and were manufactured....

l113 of was 0%???

l121- correct to: Five chickens per replicate

l145- correct to: Five chickens per replicate

l347: you say "could increase". But your results showed that it actually increased villus height or what??

l358 and l360: these are the same "cytokines play an important role in the immune system" why the repetition?

l361: AMP? where have you explained this abbreviation in the text

l380: no year in the reference

l385: no year in the reference

l398: enhance (because you said earlier could significantly increase.. so could significantly enhance....)

l400: which section are you talking about? where did they find lactobacillus... etc..

l411 to 417: that is a paragraph long sentence... Make it shorter please by dividing it.

l421: villus length is not morphology?

l423: intestinal morphology? what is the difference between intestinal morphology and villus length?

l426: you investigated several parameters to study the effects (or mechanism) of the additive on the broiler chickens. What do you mean by "identify specific mechanism" of MPX?

Point 34 in cover letter: where is this mentioned in the text? "sample was chosen randomly" and the person performing the analysis was not aware of the treatment..."

Point 36 in cover letter: you state that ceacal contents are part of the intestinal composition. I am not sure what you mean here. Intestinal composition (duodenum, jejunum and ileum) microbiota is very different to the caecal microbiota composition... So what do you mean "are part of the intestinal composition"?

Point 39 in cover letter: the two sentences are complete opposite. Now, you corrected it. It makes sense. You started by saying what we know about the anti.micro.pep., and then what we do not know, i.e. growth performance, then you moved to what you found in your study. This is more logical. Before, it was "best susbtitute, then you jump to unknown". How did you know it is best substitue if it is unknown?? 
